

# Impact of breast surgery on survival of patients with stage IV breast cancer: a SEER population-based propensity score matching analysis

Yuxiang Lin[1,*], Kaiyan Huang[1,*], Qiang Zeng[2], Jie Zhang[1] and Chuangui Song[1]

[1] Department of Breast Surgery, Fujian Medical University Union Hospital, Fuzhou, Fujian, China
[2] Department of Pathology, Fujian Medical University First Affiliated Hospital, Fuzhou, Fujian, China
[*] These authors contributed equally to this work.

Corresponding authors
Jie Zhang, zjie1979@gmail.com
Chuangui Song, scgfjxh@outlook.com

## ABSTRACT

**Background**. Breast surgery for stage IV breast cancer remains controversial. The aim of this study was to investigate the impact of breast surgery on survival of stage IV breast cancer patients based on the Surveillance, Epidemiology, and End Results (SEER) database from 2010 to 2015.

**Methods**. In total, 13,034 patients were selected and divided into surgery and non-surgery groups. Univariate and multivariable analyses were conducted to determine factors related to survival. Propensity score matching method was utilized to achieve balanced covariates across different groups. One-to-one (1:1) PSM was conducted to construct a matched sample consisting of pairs of surgery and non-surgery subjects. Breast cancer-specific survival (BCSS) and overall survival (OS) of the two groups were assessed by Kaplan–Meier plots and Cox proportional hazard regression models. Stratified analysis according to different variables was also performed.

**Results**. Patients treated with breast surgery were more likely to be younger, smaller tumor size, more advanced nodal status, higher histology grade and higher proportion of bone-only metastasis. Those who received chemotherapy and radiotherapy also tended to be treated with surgery. After adjustment for potential confounders, breast surgery group exhibited a better survival both in BCSS (HR = 0.557, 95% CI [0.523–0.594], $p < 0.001$) and OS (HR = 0.571, 95% CI [0.537–0.607], $p < 0.001$). After propensity score matching, the surgery and non-surgery group consisted of 2,269 patients respectively. The median survival time was 43 months for the surgery group and 27 months for the non-surgery group. Kaplan–Meier curves indicated that breast surgery could clearly improve both the BCSS and OS for patients with stage IV breast cancer. On multivariate analysis, surgery group was associated with a better survival compared with the non-surgery group (BCSS: HR = 0.542, 95% CI [0.499–0.589], $p < 0.001$; OS: HR = 0.555, 95% CI [0.512–0.601], $p < 0.001$). Furthermore, this survival advantage persisted in all subgroups irrespective of age, race, tumor size, nodal status, histology grade, molecular subtype, chemotherapy status, radiotherapy status or status of distant metastasis.

**Conclusion**. Our study provided additional evidence that patients with stage IV breast cancer could benefit from breast surgery and it might play a more important role in multimodality therapy.

## INTRODUCTION

Stage IV breast cancer (BC) refers to the tumor which has disseminated to distant sites. It is estimated that 3–25% of female breast cancer patients might have metastatic disease at presentation (*Siegel, Miller & Jemal, 2019*; *Miller et al., 2017*; *Eng et al., 2016a*; *Unger-Saldaña, 2014*).

The main purpose of treatment for de novo stage IV breast cancer is to alleviate symptoms, improve the quality of life and prolong survival (*Sanchez-Munoz et al., 2008*). Advances in systemic treatment have greatly improved the control of metastatic disease and prolonged survival (*Dawood et al., 2015*; *Corona et al., 2017*). Therefore, the utility of breast surgery has become a question worth discussing. Several retrospective studies have demonstrated that local surgery was associated with a better survival in women with metastatic breast cancer (*AlJohani et al., 2016*; *Warschkow et al., 2016*; *Thomas et al., 2016*; *Arciero et al., 2019*) or in specific subgroups (*Rashaan et al., 2012*). However, results from three prospective randomized trials have revealed discordant results with conflicting data (*Fitzal et al., 2019*; *Soran et al., 2018*; *Badwe et al., 2015*). In addition, it is noted that the act of surgery might accelerate metastatic growth and have an adverse effect on survival (*Gunduz, Fisher & Saffer, 1979*; *Al-Sahaf et al., 2010*; *Retsky et al., 2004*). Therefore, most guidelines still recommend surgical intervention in palliative situations or selected patients after response to initial systemic therapy.

To date, the role of surgery for de novo stage IV breast cancer patients is still ambiguous and no consensus exists. Accordingly, we conducted this propensity score matching analysis to investigate the impact of breast surgery on survival of stage IV breast cancer patients with data from a large population-based database (the Surveillance, Epidemiology, and End Results, SEER) collected from 2010 to 2015.

## MATERIAL AND METHODS

### Study patients

We performed a retrospective study of women with an initial diagnosis of stage IV breast cancer who were recorded in the SEER*Stat version 8.3.4 database from 2010 to 2015 to ensure complete data and adequate follow-up duration. SEER database collects and publishes cancer incidence and survival data from population-based cancer registries covering about 30% of the population of the United States. According to the SEER program, the initial de novo IV stage breast cancer is defined if metastases are diagnosed in the first four months after the diagnosis. In our study, we analyzed age, race, histological grade, tumor size, nodal status, breast subtype, surgery status, type of surgery, chemotherapy status, radiation status and status of distant metastasis.

All subjects who received surgical treatment related to the primary tumor (masctomy or breast conserving surgery) were included in the surgery group. Patients who did not receive any resection of their primary tumor were categorized as not having surgery. The

SEER database only provided information on radiotherapy administration (postoperative external beam radiotherapy or no radiotherapy/unknown), while it did not specify the site of radiotherapy. Therefore, the site of the surgery was the primary site (breast) and the site of the radiotherapy after the surgery could be the primary site and/or metastatic sites (such as bone). Information about the chemotherapy status was defined as "Yes" or "No/Unknown", while the regimen of chemotherapy was not supplied. In addition, the data about the anti-HER2 targeted therapy and endocrine therapy was also not specified.

The SEER database offered only the first course treatment information at the time of diagnosis and did not provide treatment information after relapse or progression. Therefore, all treatments in this study were the first course treatment after being diagnosed with stage IV breast cancer. In order to assess the effect of status of distant metastasis on survival, we divided those patients into bone-only metastasis group and visceral metastasis group, the bone-only metastasis group was defined according to the site of metastasis (bone metastasis: "Yes"; brain, liver or lung metastasis: "No").

## Statistical analysis

Continuous variables were summarized as median (and range) and were transformed into dichotomous variables at the median value. $P$-values for comparisons of different variables were calculated by chi-squared ($\chi^2$) test or Fisher's exact test. Among women diagnosed with stage IV disease, we sought to compare the overall survival (OS) and breast cancer specific survival (BCSS) between patients who did and did not receive surgical treatment for their primary tumor. The median survival time was also calculated. Kaplan–Meier survival curves were generated to compare differences in survival probabilities over time between the surgery and non-surgery groups. Univariable and multivariable Cox regression models were used to describe the associations between surgery and risk of death. To account for large sample size, we selected the variables with $p < 0.05$ which were significantly associated with BCSS or OS in the univariable analysis. One-to-one (1:1) PSM was conducted to construct a matched sample consisting of pairs of surgery and non-surgery subjects by optimal matching algorithm. Variables that were significantly different between the two groups were utilized to generate propensity scores. Specifically, we also conducted a stratified analysis with respect to BCSS and OS by age, race, tumor size, nodal status, grade, molecular subtype, chemotherapy status, radiotherapy status and bone-only metastasis or no. Psmatch2 module was used to perform propensity score matching in Stata version 13.0 (SAS Institute Inc., Cary, NC, USA). Other statistical analyses were performed with Statistical Package for the Social Sciences (SPSS, version 24.0) for Windows (IBM, Chicago, IL, USA), with a two-sided $P$ value of less than 0.05 considered statistically significant.

## RESULTS

### Clinicopathological characteristics of the selected patients

In total, 13,034 patients with a diagnosis of stage IV breast cancer between 2010–2015 who had complete information of breast surgery were included in this study. The patient selection flow-chart was displayed in Fig. 1. As shown in Table 1, 9,151 (70.2%) patients did not receive surgery and 3,883 (29.8%) were treated with surgery. There were significant

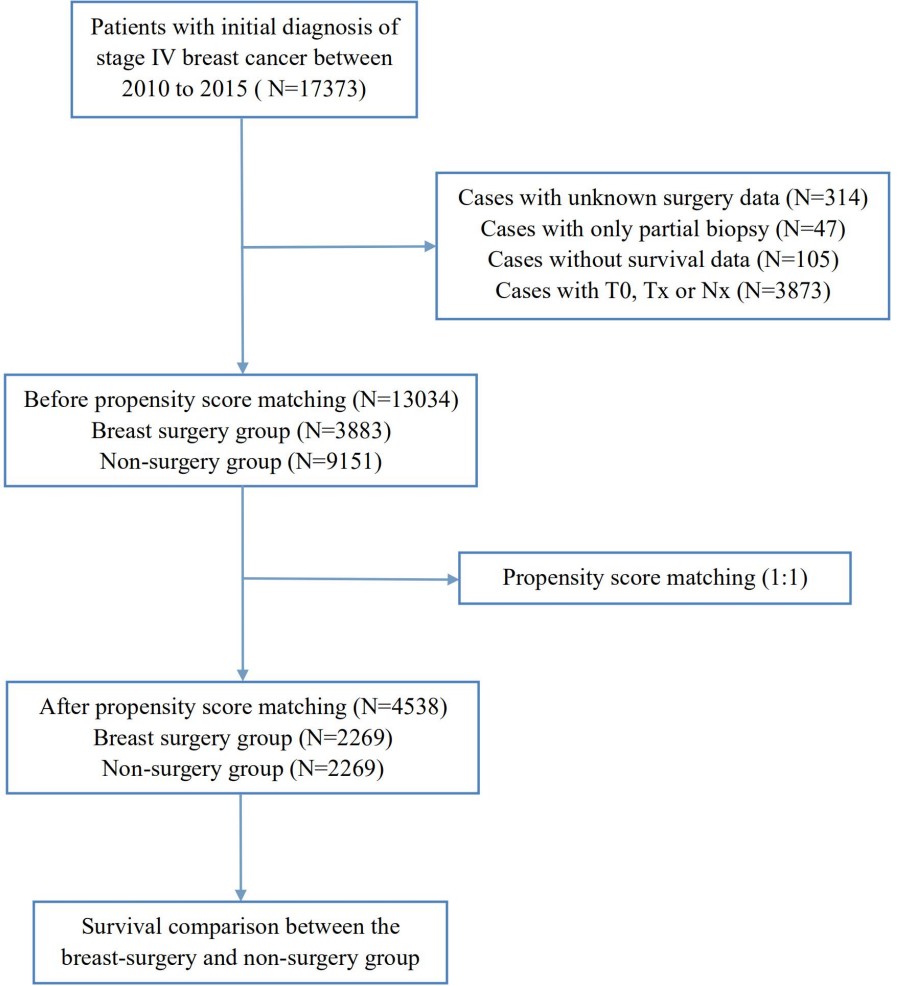

**Figure 1  Flow chart for the patient selection from SEER database.**

differences between these two groups. Patients treated with breast surgery were more likely to be younger, smaller tumor size, more advanced nodal status, higher histology grade and higher proportion of bone-only metastasis. Furthermore, those who received chemotherapy and radiotherapy also tended to be treated with surgery.

## Comparison of survival between the surgery and non-surgery groups in all patients

After the baseline characteristics were summarized, we used the Cox proportional hazards model to investigate the effect of baseline characteristics on survival outcomes. A univariable analysis indicated that older age, more advanced T or N stage, higher histology grades, triple negative breast cancer, visceral metastasis, an absence of chemotherapy or radiotherapy and patients without surgery were significantly associated with a worse BCSS and OS ($p < 0.001$) (Table S1). Furthermore, we included all variables mentioned earlier in the multivariable analysis. After adjustment for potential confounders, breast surgery was identified as an

**Table 1  Baseline Characteristics of stage IV patients with or without breast surgery before and after propensity score matching (PSM).**

| Characteristics | Before PSM | | | | $P^a$ | After PSM | | | | $P^a$ |
|---|---|---|---|---|---|---|---|---|---|---|
| | Surgery (n = 3,883) | | Non-Surgery (n = 9,151) | | | Surgery (n = 2,269) | | Non-Surgery (n = 2,269) | | |
| | No | % | No | % | | No | % | No | % | |
| Age (years) | | | | | | | | | | |
| 20–49 | 1,163 | 30.0 | 1,836 | 20.1 | <0.001 | 604 | 26.6 | 600 | 26.4 | 0.893 |
| 50–79 | 2,720 | 70.0 | 7,315 | 79.9 | | 1,665 | 73.4 | 1,669 | 73.6 | |
| Race | | | | | | | | | | |
| White | 2,841 | 73.2 | 6,747 | 73.7 | 0.012 | 1,643 | 72.4 | 1,641 | 72.3 | 0.917 |
| Black | 685 | 17.6 | 1,646 | 18.0 | | 418 | 18.4 | 431 | 19.0 | |
| Others | 349 | 9.0 | 712 | 7.8 | | 204 | 9.0 | 193 | 8.5 | |
| Unknown | 8 | 0.2 | 46 | 0.5 | | 4 | 0.2 | 4 | 0.2 | |
| T stage | | | | | | | | | | |
| T1 + T2 | 1,903 | 49.0 | 3,675 | 40.2 | <0.001 | 1,041 | 45.9 | 1,059 | 46.7 | 0.592 |
| T3 + T4 | 1,980 | 51.0 | 5,476 | 59.8 | | 1,228 | 54.1 | 1,210 | 53.3 | |
| N stage | | | | | | | | | | |
| N0 + N1 | 2,167 | 55.8 | 7,201 | 78.7 | <0.001 | 1,426 | 62.8 | 1,458 | 64.3 | 0.324 |
| N2 + N3 | 1,716 | 44.2 | 1,950 | 21.3 | | 843 | 37.2 | 811 | 35.7 | |
| Grade | | | | | | | | | | |
| I + II | 1,314 | 33.8 | 4,314 | 47.1 | <0.001 | 827 | 36.5 | 847 | 37.3 | 0.196 |
| III | 2,157 | 55.6 | 3,860 | 42.2 | | 1,287 | 56.7 | 1,296 | 57.1 | |
| Unknown | 412 | 10.6 | 977 | 10.7 | | 155 | 6.8 | 126 | 5.6 | |
| Histology | | | | | | | | | | |
| IDC | 2,973 | 76.6 | 5,745 | 62.8 | <0.001 | 1,673 | 73.7 | 1,694 | 74.7 | 0.343 |
| ILC | 286 | 7.4 | 868 | 9.5 | | 184 | 8.1 | 158 | 7.0 | |
| Others | 624 | 16.0 | 2,538 | 27.7 | | 412 | 18.2 | 417 | 18.3 | |
| Molecular subtype | | | | | | | | | | |
| HR+/HER− | 1,898 | 48.9 | 4,524 | 49.4 | <0.001 | 1,085 | 47.8 | 1,075 | 47.4 | 0.663 |
| HR+/HER− | 662 | 17.0 | 1,320 | 14.4 | | 387 | 17.1 | 393 | 17.3 | |
| HR −/HER+ | 416 | 10.7 | 716 | 7.8 | | 229 | 10.1 | 251 | 11.0 | |
| TNBC | 652 | 16.8 | 956 | 10.5 | | 413 | 18.2 | 385 | 17.0 | |
| Unknown | 255 | 6.6 | 1,635 | 17.9 | | 155 | 6.8 | 165 | 7.3 | |
| Chemotherapy status | | | | | | | | | | |
| Yes | 2,875 | 74.0 | 4,587 | 50.1 | <0.001 | 1,545 | 68.1 | 1,532 | 67.5 | 0.680 |
| No/Unknown | 1,008 | 26.0 | 4,564 | 49.9 | | 724 | 31.9 | 737 | 32.5 | |
| Radiation status | | | | | | | | | | |
| Yes | 1,802 | 46.4 | 540 | 5.9 | <0.001 | 405 | 17.8 | 438 | 19.3 | 0.208 |
| No/Unknown | 2,081 | 53.6 | 8,611 | 94.1 | | 1,864 | 82.2 | 1,831 | 80.7 | |
| Bone only metastasis | | | | | | | | | | |
| Yes | 1,506 | 38.8 | 2,971 | 32.5 | <0.001 | 768 | 33.8 | 780 | 34.4 | 0.707 |
| No | 2,377 | 61.2 | 6,180 | 67.5 | | 1,501 | 66.2 | 1,489 | 65.6 | |

**Notes.**

Abbreviations: PSM, propensity-score matching; HR, hormone receptor; HER2, human epidermal growth factor receptor 2; TNBC, triple negative breast cancer.

[a]The P value was calculated among all groups by the Chi-square test.

independent protective factor for both BCSS (HR = 0.557, 95% CI [0.523–0.594], $p<0.001$) and OS (HR = 0.571, 95% CI [0.537–0.607], $p<0.001$) (Table 2).

## Survival analysis in matched groups

To further evaluate the detected differences between breast surgery and non-surgery groups, we performed a 1:1 matched case-control analysis using the propensity score matching method. Propensity score matching between the surgery and non-surgery groups was conducted by all variables (age, race, T and N categories, histology, grade, molecular subtype, chemotherapy or radiation status, bone-only metastasis or not). After PSM, the surgery and non-surgery group consisted of 2,269 patients respectively. No statistical differences were observed between the two groups. Kaplan–Meier curves of the BCSS and OS in the surgery and non-surgery groups after PSM are presented in Fig. 2. Breast surgery clearly improved both the BCSS and OS for patients with de novo stage IV breast cancer. The median survival time was 43 months for the surgery group with 27 months for the non-surgery group. Univariable and multivariate analysis using the Cox proportional hazards model was also performed, with the relevant results shown in Table S2 and Table 3. As expected, the surgery group was associated with a marked survival advantage compared with the non-surgery group (BCSS: HR = 0.542, 95% CI [0.499–0.589], $p<0.001$; OS: HR = 0.555, 95% CI [0.512–0.601], $p < 0.001$).

## Stratified survival analysis

Furthermore, we performed a stratified analysis according to different variables in 1:1 matched groups. The Kaplan–Meier survival function was used to generate Figs. 3 and 4 in the hierarchical analysis, which represent the overall survival between surgery and non-surgery patients with different tumor size, nodal status, molecular subtypes and status of distant metastases. The median survival time for hormone receptor positive HER2 negative (HR + HER2-) and triple negative (TNBC) subtype was 47 months (surgery) vs. 32 months (non-surgery) and 16 months (surgery) vs. 11 months (non-surgery) respectively. While for bone-only metastasis and visceral metastasis patients, the median survival time was 52 months (surgery) vs. 36 months (non-surgery) and 36 months (surgery) vs. 22 months (non-surgery) respectively. Table 4 shows the hazards ratio (HR) and 95% confidence interval (CI) of the surgery group, which was determined by Cox regression analysis contrasted with that of the non-surgery group. Breast surgery was indicated to significantly reduce mortality risk regardless of tumor size, nodal status, molecular subtype or status of distant metastasis. Similarly in other subgroups (Fig. 5), surgery also presented a more favorable overall survival irrespective of age, race, histology grade chemotherapy status or radiotherapy status.

## Multivariate analysis for patients in the surgery group

We also performed a multivariate analysis by Cox proportional hazards model for patients with breast surgery in 1:1 matched groups (Table 5). For both BCSS and OS, older age, more advanced T stage, higher histology grades, triple negative breast cancer, visceral metastasis and an absence of chemotherapy presented a worse prognosis. N stage and type of surgery (masctomy or breast conserving surgery) remained irrelevant to the survival

**Table 2  Multivariate Cox proportional hazard model for breast cancer-specific survival (BCSS) and overall survival (OS) in all patients with stage IV breast cancer.**

| Variables | n | BCSS | | OS | |
|---|---|---|---|---|---|
| | | HR (95% CI) | P[a] | HR (95% CI) | P[a] |
| Age (years) | | | | | |
| 20–49 | 2,999 | Reference | | Reference | |
| 50–79 | 1,0035 | 1.282 (1.209–1.359) | <0.001 | 1.318 (1.245–1.396) | <0.001 |
| Race | | | | | |
| White | 9,588 | Reference | | Reference | |
| Black | 2,331 | 1.212 (1.143–1.285) | <0.001 | 1.241 (1.173–1.312) | <0.001 |
| Others | 1,061 | 0.966 (0.885–1.055) | 0.444 | 0.941 (0.863–1.026) | 0.166 |
| Unknown | 54 | 0.255 (0.133–0.491) | <0.001 | 0.237 (0.123–0.456) | <0.001 |
| T stage | | | | | |
| T1 + T2 | 5,578 | Reference | | Reference | |
| T3 + T4 | 7,456 | 1.239 (1.181–1.299) | <0.001 | 1.234 (1.178–1.292) | <0.001 |
| N stage | | | | | |
| N0 + N1 | 9,368 | Reference | | Reference | |
| N2 + N3 | 3,666 | 1.015 (0.963–1.070) | 0.584 | 1.017 (0.966–1.070) | 0.526 |
| Grade | | | | | |
| I + II | 5,628 | Reference | | Reference | |
| III + IV | 6,017 | 1.333 (1.265–1.405) | <0.001 | 1.305 (1.240–1.373) | <0.001 |
| Unknown | 1,389 | 1.031 (0.951–1.118) | 0.461 | 1.028 (0.950–1.111) | 0.494 |
| Histology | | | | | |
| IDC | 8,718 | Reference | | Reference | |
| ILC | 1,154 | 1.083 (0.994–1.181) | 0.067 | 1.058 (0.973–1.149) | 0.188 |
| Others | 3,162 | 1.178 (1.114–1.246) | <0.001 | 1.191 (1.129–1.257) | <0.001 |
| Molecular subtype | | | | | |
| HR+/HER2− | 6,422 | Reference | | Reference | |
| HR+/HER2+ | 1,982 | 0.825 (0.763–0.892) | <0.001 | 0.821 (0.762–0.886) | <0.001 |
| HR−/HER2+ | 1,132 | 1.005 (0.915–1.105) | 0.912 | 1.007 (0.919–1.104) | 0.879 |
| TNBC | 1,608 | 2.537 (2.361–2.725) | <0.001 | 2.480 (2.313–2.660) | <0.001 |
| Unknown | 1,890 | 1.573 (1.471–1.682) | <0.001 | 1.574 (1.476–1.679) | <0.001 |
| Chemotherapy status | | | | | |
| Yes | 7,462 | Reference | | Reference | |
| No/Unknown | 5,572 | 1.591 (1.511–1.676) | <0.001 | 1.618 (1.539–1.700) | <0.001 |
| Radiation status | | | | | |
| No/Unknown | 1,0692 | Reference | | Reference | |
| Yes | 2,342 | 0.900 (0.837–0.967) | 0.004 | 0.874 (0.815–0.938) | <0.001 |
| Bone-only metastasis | | | | | |
| Yes | 4,477 | Reference | | Reference | |
| No | 8,557 | 1.588 (1.507–1.674) | <0.001 | 1.566 (1.488–1.647) | <0.001 |
| Surgery status | | | | | |
| No | 9,151 | Reference | | Reference | |
| Yes | 3,883 | 0.557 (0.523–0.594) | <0.001 | 0.571 (0.537–0.607) | <0.001 |

**Notes.**

Abbreviation: HR, hazard ratio; CI, confidence interval; BCSS, breast cancer-specific survival; OS, overall survival; HR, hormone receptor; HER2, human epidermal growth factor receptor 2; TNBC, triple negative breast cancer.

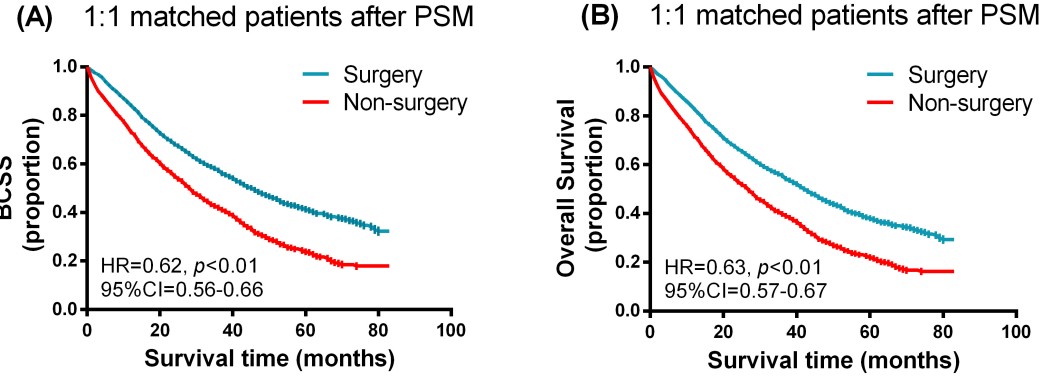

**Figure 2** Kaplan–Meier curves of breast cancer specific survival (A) and overall survival (B) in the surgery and non-surgery groups after propensity score matching.

of this group of patients. While radiotherapy was identified to be a significantly favorable factor both in BCSS and OS (HR = 0.819, 95% CI [0.694–0.966], $p = 0.018$; HR = 0.783, 95% CI [0.667–0.920], $p = 0.003$).

## DISCUSSION

In this large population-based cohort study, we sought to reveal the distinct outcomes of stage IV breast cancer with or without surgical intervention based on the SEER population-based data. Our findings indicated that the surgery group was associated with a better survival compared with the non-surgery group (BCSS: HR = 0.542, 95% CI [0.499–0.589], $p < 0.001$; OS: hR = 0.555, 95% CI [0.512–0.601], $p < 0.001$). Furthermore, this survival advantage persisted in all subgroups irrespective of age, race, tumor size, nodal status, histology grade, molecular subtype, chemotherapy status, radiotherapy status or status of distant metastasis.

Metastatic breast cancer is still considered as a systemic disease and local therapy would only have little impact on outcomes (*Lane et al., 2019*). The primary aim of treatment is to alleviate symptoms, improve the quality of life and prolong survival. In clinical practice, the majority of patients with de novo stage IV breast cancer are recommended to receive systemic therapy including chemotherapy, anti-HER2 therapy or endocrine therapy. Surgery is mainly considered when there is tumor bleeding or ulceration (*Arnedos et al., 2015*). Earlier studies also suggested that the growth of distant metastases could be stimulated by removal of primary tumor. Surgical intervention could reduce angiostatin secretion and stimulate the release of growth factors, thus accelerating metastatic growth and presenting an adverse effect on survival (*Gunduz, Fisher & Saffer, 1979*; *Al-Sahaf et al., 2010*; *Retsky et al., 2004*; *Folkman, 1996*). However, other experimental studies in the murine model indicated that substantially reducing overall tumor burden with local surgery could lead to improved survival (*Rashid et al., 2013*). This might be explained by

**Table 3  Multivariate Cox proportional hazard model for breast cancer-specific survival (BCSS) and overall survival (OS) in 1:1 matched propensity score matching analysis with stage IV breast cancer.**

| Variables | n | BCSS | | OS | |
|---|---|---|---|---|---|
| | | HR (95% CI) | P [a] | HR (95% CI) | P [a] |
| **Age (years)** | | | | | |
| 20–49 | 1,204 | Reference | | Reference | |
| 50–79 | 3,334 | 1.203 (1.095–1.303) | <0.001 | 1.257 (1.145–1.379) | <0.001 |
| **Race** | | | | | |
| White | 3,284 | Reference | | Reference | |
| Black | 849 | 1.252 (1.134–1.382) | <0.001 | 1.286 (1.169–1.415) | <0.001 |
| Others | 397 | 0.887 (0.761–1.035) | 0.127 | 0.879 (0.751–1.021) | 0.092 |
| Unknown | 8 | 0.179 (0.025–1.272) | 0.085 | 0.167 (0.024–1.189) | 0.074 |
| **T stage** | | | | | |
| T1 + T2 | 2,100 | Reference | | Reference | |
| T3 + T4 | 2,438 | 1.408 (1.294–1.531) | <0.001 | 1.407 (1.297–1.526) | <0.001 |
| **N stage** | | | | | |
| N0 + N1 | 2,884 | Reference | | Reference | |
| N2 + N3 | 1,654 | 1.091 (1.003–1.185) | 0.042 | 1.098 (1.012–1.190) | 0.024 |
| **Grade** | | | | | |
| I + II | 1,674 | Reference | | Reference | |
| III + IV | 2,583 | 1.587 (1.436–1.753) | <0.001 | 1.517 (1.378–1.670) | <0.001 |
| Unknown | 281 | 1.279 (1.118–1.464) | <0.001 | 1.242 (1.090–1.414) | 0.001 |
| **Molecular subtype** | | | | | |
| HR+/HER2− | 2,160 | Reference | | Reference | |
| HR+/HER2+ | 780 | 0.702 (0.612–0.805) | <0.001 | 0.714 (0.617–0.804) | <0.001 |
| HR −/HER2+ | 480 | 0.923 (0.790–1.078) | 0.311 | 0.936 (0.806–1.088) | 0.388 |
| TNBC | 798 | 2.663 (2.373–2.988) | <0.001 | 2.603 (2.327–2.912) | <0.001 |
| Unknown | 320 | 1.485 (1.274–1.731) | <0.001 | 1.486 (1.282–1.722) | <0.001 |
| **Chemotherapy status** | | | | | |
| Yes | 3,077 | Reference | | Reference | |
| No/Unknown | 1,461 | 1.554 (1.412–1.710) | <0.001 | 1.577 (1.438–1.729) | <0.001 |
| **Bone-only metastasis** | | | | | |
| Yes | 1,548 | Reference | | Reference | |
| No | 2,990 | 1.390 (1.266–1.525) | <0.001 | 1.369 (1.252–1.498) | <0.001 |
| **Surgery status** | | | | | |
| No | 2,269 | Reference | | Reference | |
| Yes | 2,269 | 0.542 (0.499–0.589) | <0.001 | 0.555 (0.512–0.601) | <0.001 |

**Notes.**

Abbreviation: HR, hazard ratio; CI, confidence interval; BCSS, breast cancer-specific survival; OS, overall survival; HR, hormone receptor; HER2, human epidermal growth factor receptor 2; TNBC, triple negative breast cancer.

[a] The P value was adjusted by the multivariate Cox proportional hazard regression model.

the primary tumor's suppression of the immune response and its surgical removal could result in restored immunocompetence (*Danna et al., 2004*).

Therefore, the utility of surgical intervention in this population has long been debated. Multiple retrospective studies have revealed the potential benefit with surgery (*AlJohani*

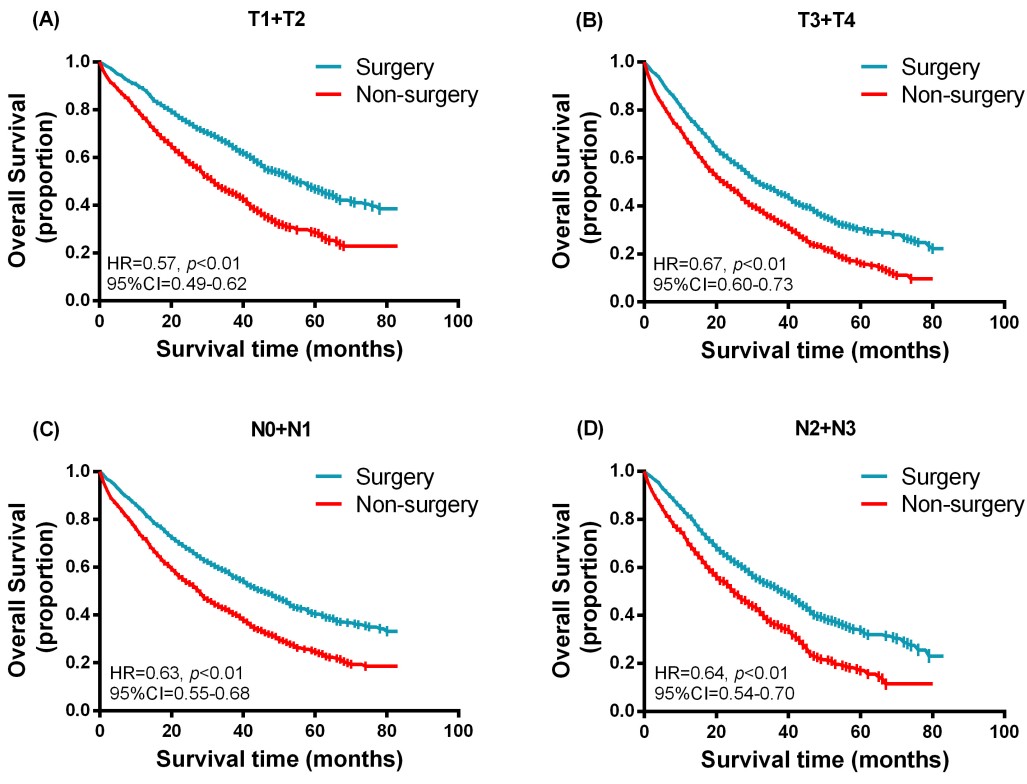

**Figure 3** **Kaplan–Meier curves of overall survival in the surgery and non-surgery groups stratified by different tumor size and nodal status.** (A) T1 + T2, (B) T3 + T4, (C) N0 + N1, (D) N2 + N3.

*et al., 2016*; *Warschkow et al., 2016*; *Thomas et al., 2016*; *Arciero et al., 2019*; *Rashaan et al., 2012*; *Gnerlich et al., 2007*; *Quinn et al., 2015*; *Leung et al., 2010*; *Blanchard et al., 2008*; *Eng et al., 2016b*). The most recent study based on the SEER database (1998–2011) proposed a survival advantage with surgical intervention (median overall survival, 34 months for surgery vs. 18 months for non-surgery) (*Vohra et al., 2018*). However, the data about HER2 status in this study were incomplete and no stratified analysis was conducted. One study based on NCDB database also noted a benefit for stage IV breast cancer patients with surgery (*Arciero et al., 2019*). In a large cohort of 11,694 patients, an improved overall survival was observed for the surgery group compared with the non-surgery group after propensity score matching (HR = 0.68, 95% CI [0.63–0.72], $p < 0.001$). These conclusions are similar to the results in our study, providing consistent evidence from registry-based retrospective studies that well-selected patients with de novo stage IV breast cancer who undergo surgical intervention could obtain a better survival.

In spite of the evidence in several retrospective studies, supportive prospective analyses still lacked. Fitzal's study (ABCSG-28 POSYTIVE) enrolled 90 previously untreated stage IV breast cancer patients and randomly assigned them to surgical resection followed by systemic therapy group or primary systemic therapy group (*Fitzal et al., 2019*). This trial was stopped early due to poor recruitment and the median overall survival for surgery and

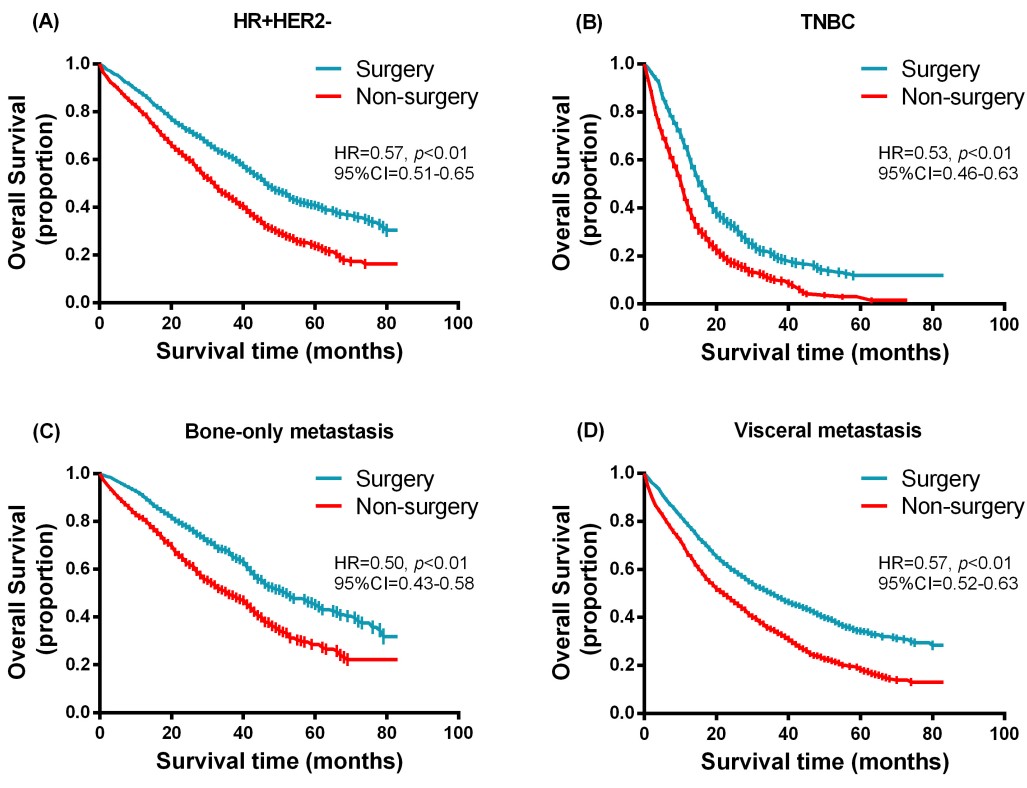

**Figure 4  Kaplan–Meier curves of overall survival in the surgery and non-surgery groups stratified by molecular subtypes and status of distant metastasis.** (A) HR+HER2-, (B) TNBC, (C) bone-only metastasis, (D) visceral metastasis.

non-surgery group was 34.6 and 54.8 months respectively (HR = 0.691, 95% CI [0.358–1.333]; $p = 0.267$). MF07-01 trial (*Soran et al., 2018*) is another prospective, multicenter, phase III, randomized trial to focus on the impact of breast surgery on the survival of de novo stage IV BC patients. In this study, one group received sequential systemic therapy after primary surgery and the other group only received systemic therapy alone. Local surgery did not gained a survival advantage after 3 years of follow-up. But after 5 years of follow-up, patients with local surgery achieved a better overall survival (HR = 0.66, 95% CI [0.49–0.88]; $p = 0.005$). Unplanned subgroup analyses indicated that the survival benefit of breast surgery presented in patients with younger age (<55 years), ER/PR positive, HER2 negative or solitary bone-only metastases. Although these findings identified the therapeutic value of breast surgery and suggested several factors such as molecular subtype or metastatic site that should be taken into consideration, controversy still existed for the procedure of surgical resection followed by systemic therapy did not accord with the clinical practice now. The other prospective trial by Badwe et al. (*Badwe et al., 2015*) randomly included 350 previously untreated de novo metastatic BC patients from India between 2005 to 2013. Median overall survival was 19.2 months (95% CI [15.98–22.46]) in the surgery group and 20.5 months (16.96–23.98) in the non-surgery group (HR = 1.04, 95% CI [0.81–1.34]; $p = 0.79$). The uncertain effect of surgery in this study might be attributed

**Table 4** Multivariate Cox proportional hazard regression model of breast cancer-specific survival (BCSS) and overall survival (OS) for the 1:1 matched surgery and non-surgery groups, stratified by the T stage, N stage, breast subtype and metastasis status.

| Variables[b] | Surgery vs. Non-surgery[a] | | | |
|---|---|---|---|---|
| | BCSS | | OS | |
| | HR (95% CI) | P | HR (95% CI) | P |
| T stage | | | | |
| T1 + T2 | 0.492 (0.431–0.562) | <0.001 | 0.504 (0.443–0.572) | <0.001 |
| T3 + T4 | 0.580 (0.521–0.646) | <0.001 | 0.594 (0.535–0.659) | <0.001 |
| N stage | | | | |
| N0 + N1 | 0.528 (0.475–0.587) | <0.001 | 0.538 (0.486–0.596) | <0.001 |
| N2 + N3 | 0.564 (0.493–0.646) | <0.001 | 0.585 (0.513–0.667) | <0.001 |
| Breast subtype | | | | |
| HR+/HER2− | 0.554 (0.489–0.628) | <0.001 | 0.573 (0.508–0.646) | <0.001 |
| HR +/HER2+ | 0.462 (0.361–0.592) | <0.001 | 0.473 (0.372–0.601) | <0.001 |
| HR −/HER2+ | 0.459 (0.346–0.609) | <0.001 | 0.490 (0.374–0.643) | <0.001 |
| TNBC | 0.536 (0.455–0.631) | <0.001 | 0.534 (0.455–0.627) | <0.001 |
| Metastasis status | | | | |
| Bone-only metastasis | 0.495 (0.423–0.580) | <0.001 | 0.501 (0.431–0.583) | <0.001 |
| Visceral metastasis | 0.562 (0.510–0.619) | <0.001 | 0.568 (0.517–0.625) | <0.001 |

**Notes.**

[a] Non-surgery as a reference.

[b] Adjusted by a multivariate Cox proportional model, including age, race, T stage, N stage, grade, molecular subtype, chemotherapy status, solitary bone or visceral metastasis where appropriate.

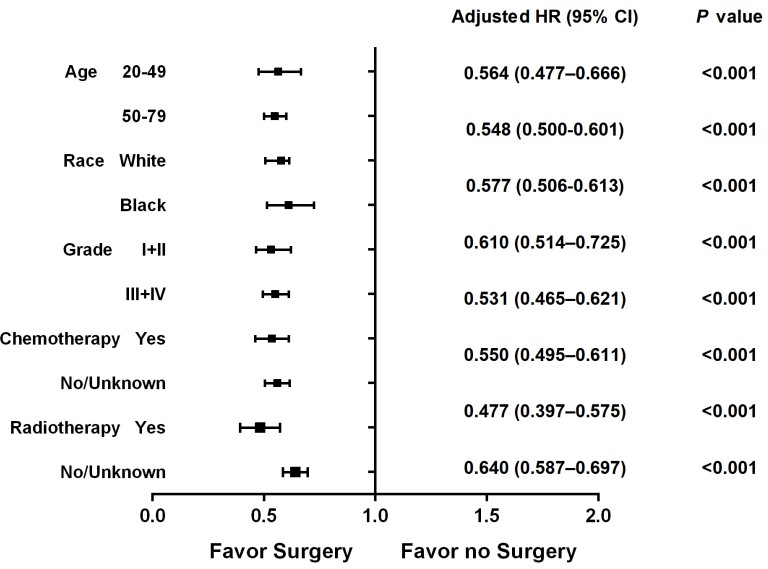

**Figure 5** Forest plot of overall survival in the surgery and non-surgery groups stratified by age, race, histology grade, chemotherapy status and radiotherapy status.

**Table 5** Multivariate analyses for breast cancer-specific survival (BCSS) and overall survival (OS) in stage IV breast cancer patients with breast surgery in the 1:1 matched groups.

| Variables | n | BCSS | | OS | |
|---|---|---|---|---|---|
| | | HR (95% CI) | P[a] | HR (95% CI) | P[a] |
| Age (years) | | | | | |
| 20–49 | 604 | Reference | | Reference | |
| 50–79 | 1,665 | 1.171 (1.017–1.348) | 0.028 | 1.240 (1.081–1.423) | 0.002 |
| Race | | | | | |
| White | 1,643 | Reference | | Reference | |
| Black | 418 | 1.294 (1.115–1.501) | <0.001 | 1.372 (1.191–1.581) | <0.001 |
| Others | 204 | 0.763 (0.662–0.954) | 0.018 | 0.762 (0.614–0.947) | 0.014 |
| Unknown | 4 | NA | | NA | |
| T stage | | | | | |
| T1 + T2 | 1,041 | Reference | | Reference | |
| T3 + T4 | 1,228 | 1.572 (1.382–1.787) | <0.001 | 1.568 (1.386–1.774) | <0.001 |
| N stage | | | | | |
| N0 + N1 | 1,426 | Reference | | Reference | |
| N2 + N3 | 843 | 1.106 (0.975–1.255) | 0.118 | 1.113 (0.985–1.257) | 0.085 |
| Grade | | | | | |
| I + II | 827 | Reference | | Reference | |
| III + IV | 1,287 | 1.625 (1.399–1.887) | <0.001 | 1.514 (1.314–1.745) | <0.001 |
| Unknown | 155 | 1.316 (1.020–1.697) | 0.034 | 1.250 (0.978–1.597) | 0.075 |
| Molecular subtype | | | | | |
| HR+/HER2− | 1,085 | Reference | | Reference | |
| HR+/HER2+ | 387 | 0.580 (0.468–0.719) | <0.001 | 0.585 (0.476–0.719) | <0.001 |
| HR−/HER2+ | 229 | 0.765 (0.599–0.978) | 0.033 | 0.791 (0.626–0.999) | 0.049 |
| TNBC | 413 | 2.486 (2.105–2.936) | <0.001 | 2.392 (2.036–2.812) | <0.001 |
| Unknown | 155 | 1.507 (1.208–1.879) | <0.001 | 1.512 (1.224–1.868) | <0.001 |
| Type of surgery | | | | | |
| BCS | 629 | Reference | | Reference | |
| Masctomy | 1,640 | 1.105 (0.965–1.267) | 0.149 | 1.032 (0.913–1.235) | 0.187 |
| Chemotherapy status | | | | | |
| Yes | 1,545 | Reference | | Reference | |
| No/Unknown | 724 | 1.500 (1.302–1.730) | <0.001 | 1.531 (1.336–1.754) | <0.001 |
| Bone-only metastasis | | | | | |
| Yes | 768 | Reference | | Reference | |
| No | 1,501 | 1.368 (1.189–1.594) | <0.001 | 1.370 (1.198–1.568) | <0.001 |
| Radiation status | | | | | |
| No/Unknown | 405 | Reference | | Reference | |
| Yes | 1,864 | 0.819 (0.694–0.966) | 0.018 | 0.783 (0.667–0.920) | 0.003 |

**Notes.**

Abbreviation: HR, hazard ratio; CI, confidence interval; BCSS, breast cancer-specific survival; OS, overall survival; HR, hormone receptor; HER2, human epidermal growth factor receptor 2; TNBC, triple negative breast cancer; BCS, breast conserving surgery.

[a] The P value was adjusted by the multivariate Cox proportional hazard regression model.

to the fact that only few patients enrolled received paclitaxel-based chemotherapy and most of HER2 positive patients did not take anti-HER2 therapy.

Our current study of the SEER database provided strong retrospective data of breast surgery in stage IV breast cancer. It is expected that patients with lower disease burden and better prognostic factors such as ER+HER- subtype or bone-only metastasis are more likely to undertake surgery, thereby resulting a better prognosis. In a matched paired retrospective analysis, it is noted that selection bias in stage IV breast cancer could affect the survival outcomes (*Cady et al., 2008*). Therefore, propensity score matching analysis was applied in our study to balance covariates in different groups and reduce selection bias. The results of propensity score matching indicated that surgical intervention obtained a significant survival benefit. Furthermore, patients with surgery were shown to significantly reduce mortality risk in different subgroups, regardless of age, race, histology grade, tumor size, nodal status, molecular subtype, chemotherapy status or status of distant metastasis, suggesting that breast surgery might have independent therapeutic value to improve survival in stage IV breast cancer. However, one point that should be mentioned is a relatively poor survival for stage IV triple negative breast cancer (TNBC) patients. The median survival time for TNBC patients was 16 months (surgery) vs. 11 months (non-surgery) respectively. Although surgical intervention revealed a better survival outcome, whether these patients should received surgery required further discussion. For patients with breast surgery, we also performed a multivariate analysis. Type of surgery (masctomy or breast conserving surgery) remained irrelevant to the survival, while radiotherapy was identified to be a significantly favorable factor both in BCSS and OS (HR = 0.819, 95% CI [0.694–0.966], $p$ = 0.018; HR = 0.783, 95% CI [0.667–0.920], $p$ = 0.003).

Stage IV breast cancer is a group of highly heterogeneous disease. Advances in systemic treatment have greatly improved the control of metastases disease. Five-year disease special survival of de novo breast cancer has been improved from 28% (1990–1998) to 55% (2005–2010) (*Malmgren et al., 2018*). Therefore, local treatment might play a more important role than conventionally considered in metastatic breast cancer patients. However, several limitations should also be mentioned in our study. Firstly, although propensity score matching analysis was utilized, selection bias (regarding the retrospective design) and guarantee time bias (those who do not live long enough to undergo surgery are classified to the no-surgery group) still existed. Secondly, information about anti-HER2 targeted therapy and endocrine therapy is absent, while the regimen of chemotherapy and the exact site of radiotherapy (primary tumor or metastasis site such as bone) are also unavailable from the SEER database. Data on performance status and comorbidities are also not presented for they are vital as these could be the factors that prevailed in surgery decision-making process. Thirdly, the status of disease burden is incomplete (SEER database does not provide the number of metastases, but only with the information of the major sites, such as bone, lung, liver, brain and distant lymph nodes). In addition, a possibility of late diagnosis of metastatic disease could also impact the interpretation of the results (For in SEER database, the initial de novo breast cancer is defined if metastases are diagnosed in the first four months after the diagnosis). Lastly, we could not determine

the timing of surgery for patients included, whether the breast surgery was performed after systemic treatment or at initial diagnosis is also unknown.

## CONCLUSION

Our study provided additional evidence that patients with stage IV breast cancer could benefit from surgical treatment. Future multicenter, large-scale prospective studies with long-term follow-up are still warranted.

### Funding

This study was supported by grants from National Natural Science Foundation of China (81672817), Joint Funds for the Innovation of Science and Technology, Fujian Province (2017Y9033). The funders had no role in study design, data collection and analysis, decision to publish, or preparation of the manuscript.

### Grant Disclosures

The following grant information was disclosed by the authors:
National Natural Science Foundation of China: 81672817.
Joint Funds for the Innovation of Science and Technology.
Fujian Province: 2017Y9033.

### Competing Interests

The authors declare there are no competing interests.

### Author Contributions

- Yuxiang Lin performed the experiments, analyzed the data, prepared figures and/or tables, authored or reviewed drafts of the paper, and approved the final draft.
- Kaiyan Huang and Qiang Zeng performed the experiments, analyzed the data, prepared figures and/or tables, and approved the final draft.
- Jie Zhang and Chuangui Song conceived and designed the experiments, authored or reviewed drafts of the paper, and approved the final draft.

### Data Availability

The raw measurements are available in the Supplementary Files.

### Supplemental Information

Supplemental information for this article can be found online at http://dx.doi.org/10.7717/peerj.8694#supplemental-information.

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
