# Peer review of "Impact of breast surgery on survival of patients with stage IV breast cancer: a SEER population-based propensity score matching analysis"

_PeerJ, doi:10.7717/peerj.8694_

## Round 0.1 · original submission · Major Revisions

Your manuscript has been reviewed and requires modifications prior to making a decision. The comments of the reviewers are included at the bottom of this letter. Reviewer 1 indicated that the statistical methods section should be improved. Review 1 also recommended adding new references. Reviewer 2 mentioned the limitations of the study. The authors may prefer to discuss these issues in the discussion section. I agree with the evaluation and I would, therefore, request for the manuscript to be revised accordingly. I would also like to suggest the following change:

Please correct the abbreviations of Table 1. MST and IQR were not used in Table. Add IBM to SPSS software.

·

Basic reporting

This retrospective study analysed impact of breast surgery on outcomes in de novo metastatic breast cancer patients using SEER derived data. The research question is relevant, however, the appropriateness of this research approach with using propensity score matching method and SEER-derived data in this setting is questionable. Nevertheless, numerous similar studies have been performed and published beforehand. The authors are to be congratulated for providing complete raw data. However, the methods could be more precisely described.

There are several major issues that need to be considered before potential publication. Please also refer to the comments made in the peerj-reviewing-41899-v0.pdf.

1. Language: It is written in professional English that is in parts unclear mostly due to poor syntax, and redundancy, less because of ambiguousness. Please refer to in-line comments which are far from thorough but are meant to emphasize the problem which needs to be tackled.

2. Title “Impact of surgical treatment on survival for patients with stage IV breast cancer”: Authors should consider using "breast surgery" instead of "surgical treatment", as, for example, metastasectomy is also a type of surgical treatment but is not a subject of this research. This should be corrected throughout the manuscript.

3. In the Introduction the authors grasp the complexity of the research question well and they do reference the pertinent studies, however, besides minor language slips some other corrections are needed as well:

a. Line 50: "…has been transferred…" is an expression used to describe an active transfer, whereas in this case consider something along the line "has disseminated to distant sites" or similar. The authors could as well consider the whole first sentence redundant.

b. Line 50: “…to the site away from the breast…” Imprecise. For example, regional nodes are a site away from the breast but N+ disease is not considered stage IV on its own. The authors should recheck and clarify this.

c. Line 51: “…5-10% of female…” Considering the references 1-3 provided by the authors, the percentages are 4.4-6% and are valid for USA only, whereas proportion of stage IV breast cancer at diagnosis worldwide ranges from 3 to 25% according to Unger-Saldaña et al (1). The authors should recheck and clarify this.

d. Line 54-56: The authors should support this statement with pertinent reference. Or if this is their general observation from their practice, they should make a more general or/and clear statement.

e. Line 60: References [5-10]: Two of these studies (10. Lim et al, 5. Rashaan et al) failed to demonstrate a statistically significant positive impact of breast surgery on overall survival in the studied cohort. The positive effect was evident only in subgroups of patients. The authors should either correct the statement or discuss these two studies separately.

4. Materials and methods:

a. The authors used propensity score matching method (PSM). According to Biondi-Zoccai et al (2) the decision between using PSM or standard multivariable analysis is not as straightforward as it seems when it comes to retrospective analysis. One of the key factors in this decision-making process is the number of events per variable, yet nowhere in this article can the number of events be found. As this is crucial for declaring this article statistically sound, I would like to hear from the authors why they chose PSM over, for example, Cox proportional hazard analysis. Furthermore, the number of events should be provided by the authors in their paper.

b. The authors should provide the reader with the SEER definition of "de novo metastatic breast cancer". According to SEER the initial stage is defined as metastatic if metastases are diagnosed in the first four months after the diagnosis (3). This definition is crucial for understanding the limitations of retrospective studies assessing the effect of breast surgery on survival in this setting.

c. The authors should provide the reader with the patient selection flow-chart describing the patient selection process with number of initially selected patients (all C50 in SEER in 2010-2015) and so on. Furthermore, the authors should provide the reader with the basic information about SEER which also partially describes the studied cohort, e.g. "SEER collects and publishes cancer incidence and survival data from population-based cancer registries covering approximately (28%) of the population of the United States."

d. Why did the authors feel the need to dichotomize the continuous variable age? According to Cohen et al (4) this leads to a reduction in statistical power and potentially also to misleading results (5). Could the authors at least categorize the age into 10-year-intervals and redo the analyses? Alternatively, could the authors provide the reader with the reasoning behind dichotomization and the cut-off value used?

e. The authors should provide the reader with more information on what data regarding radiation was available to them from the SEER database, but not only in Discussion, but also here in Material and Methods. Line 75: Is this radiation to the primary site? To the regional nodes? To the metastatic site? Before surgery? After surgery? Before systemic therapy? After systemic therapy? This definition is crucial for understanding the limitations of the retrospective studies assessing the effect of breast surgery on survival in this setting.

f. Line 75, “chemotherapy”: As is evident from the supplementary material, "no" and "unknown" were considered together in analyses. I am very interested in why did the authors not include the chemotherapy status as "yes", "no", and "unknown" separately in the analysis as they did with other covariates?

g. Was the information on targeted treatment available to authors? If so, the authors should include this data in their analyses as non-receipt of targeted treatment in HER-2 positive patients could substantially influence the results. If not, the authors should state this crucial information already in Materials and Methods section, not only in Discussion. This was also one of the limitations in the prospective trial of Badwe et al (6) where only 9 out of 107 HER2 positive patients received anti-HER2 therapy.

h. Solitary bone metastasis. The SEER (7, see p.124) does not collect data on the number of metastases, therefore the authors could not have "divided those patients into solitary bone metastasis group and non-solitary bone metastasis". Did the authors mean to say "bone-only metastases" instead? This should be explained, rechecked and corrected accordingly throughout the text. This distinction is vital.

i. Line 92-93: “…patients who did not receive any formal resection…” Is it possible that some of the patients received "informal resection"? If so, the authors should elaborate on this. If not, the authors should reword by leaving out "formal".

j. Lines 101-103: Was delivery of radiotherapy not included in the analyses?

k. Line 101, “…survival risk…”: Further below the authors provide the readers with hazard ratios of death. Could it be more appropriate to use "risk of death" instead.

5. Results:

a. Line 114-115, “…worse histology grade…”: "higher grade" would suffice.

b. Line 127: Why is radiotherapy not included in the analysis?

c. Line 146-147: Why is radiotherapy not included in the analysis? What about chemotherapy?

d. In the titles of the tables the PSM should be worded in full.

e. Kaplan-Meier curves: Several authors including Rich et al (8) stressed the importance of showing censored patients as tick marks in survival curves. Authors should consider adding these or adding the “number at risk” below the Y-axis. This should be corrected throughout the figures.

6. In the Discussion the authors reference the pertinent studies, however, besides several language slips some other corrections are needed as well:

a. Line 166: The authors should rephrase this sentence, because metastatic breast cancer is still considered a systemic disease. The authors should rethink about the reference supporting this statement, because in reference 17 the authors performed a similar analysis with similar results in 2007 on 1988–2003 SEER data.

b. Line 176: Authors should recheck and correct their incorrect statement. Rashid et al observed in their murine model: "Surgical stress increased tumour burden only transiently without affecting survival. When primary tumour resection decreased overall tumour burden substantially, further growth of metastatic lesions did not increase overall tumour burden compared to observation and survival was improved, which was not the case when resection did not significantly reduce overall tumour burden."

c. Line 176-177: The authors should restate this, as this statement is incorrect. First, the primary tumour in situ is inducing immune suppression and not the other way around. Second, neither of these three references shows causal connection between restoring immunocompetency and improved survival. Third, the review by Demicheli et al provides us with a short review of possible mechanisms responsible for suppression of metastatic growth by the presence of primary tumour by circulating angiogenesis inhibitors as well and is as such not the most appropriate reference in this position.

d. Line 187, “evidence”: The authors should add "...evidence from registry-based retrospective studies that...", as omission of this fact could mislead the reader.

e. Line 233-234: The authors need to add a reference.

f. Limitations: For the sake of transparency, the authors should provide the reader with additional limitations:
i. Guarantee time bias: those who do not live long enough to undergo surgery are classified to the no-surgery group, leading to erroneously improved prognosis in the surgery group.
ii. A possibility of late diagnosis of metastatic disease (see above about the SEER’s 4 months to define definite initial stage) could mean that some of the patients with breast surgery were initially metastases-free or at least had metastases for a shorter period and were therefore asymptomatic at diagnosis. This could lead to a biased attribution of improved prognosis to the surgery group.
iii. Problem of residual confounding is not presented in detail as it should be. Perhaps most importantly, data on performance status and comorbidities are vital as these could be the factors that prevailed in the surgery decision-making process and are important determinants of survival.
iv. The problem of disease burden: see above - SEER does not provide us with the number of metastases, but only with the information if they are present and if they are present in any of the major sites (bones, lung, liver, brains, distant lymph nodes) at most. This is vital for the interpretation of the results, as this difference could not be discerned from the data provided by SEER, but this is a major factor in breast-surgery decision making and has an impact on survival.

g. Supplement material
i. Column heading: “ICD-O-3 Hist/behav, malignant (1=IDC,2=ILC,3=其它)”: this should be corrected.


References
1. Unger-Saldaña K. Challenges to the early diagnosis and treatment of breast cancer in developing countries. World J Clin Oncol. 2014;5(3):465.
2. Biondi-Zoccai G, Romagnoli E, Agostoni P, Capodanno D, Castagno D, D’Ascenzo F, et al. Are propensity scores really superior to standard multivariable analysis? Contemp Clin Trials. 2011 Sep;32(5):731–40.
3. Ruhl J, Adamo M, Dickie L. Section V. In: SEER Program Coding and Staging Manual 2016. National Cancer Institute, Bethesda, MD 20850-9765; 2016.
4. Jacob Cohen. The Cost of Dichotomization. Appl Psychol Meas. 1983;7(3):249–53.
5. Royston P, Altman DG, Sauerbrei W. Dichotomizing continuous predictors in multiple regression: a bad idea. Stat Med. 2006 Jan 15;25(1):127–41.
6. Badwe R, Hawaldar R, Nair N, Kaushik R, Parmar V, Siddique S, et al. Locoregional treatment versus no treatment of the primary tumour in metastatic breast cancer: an open-label randomised controlled trial. Lancet Oncol. 2015 Oct;16(13):1380–8.
7. NCISeer. SEER Program Coding and Staging Manual 2018 SEER Program Coding and Staging Manual 2018 Acknowledgements ii SEER Program Coding and Staging Manual 2018. 2018;(January).
8. Rich JT, Neely JG, Paniello RC, Voelker CCJ, Nussenbaum B, Wang EW. A practical guide to understanding Kaplan-Meier curves. Otolaryngol Neck Surg. 2010 Sep;143(3):331–6.

Experimental design

Please see 1. Basic reporting and attached files.

Validity of the findings

Please see 1. Basic reporting and attached files.

Additional comments

Please see 1. Basic reporting and attached files.

Reviewer 2 ·

Basic reporting

a. The study is written in a clear, unambiguous and professional English.
b. The authors described the background behind this topic clearly.
c. The figures and tables are relevant and of high quality.
d. The study is self-contained with relevant results to the hypothesis.

Experimental design

a. This study is not original when it comes to Ideas. However, it presented a large number of patients, which strengthen its evidential power.
b. The research questions were solid, rational and can be answered without a great deal of bias.
c. The investigatory process and statistical analysis are nearly perfect. Also, the methodology is unambiguously described.

Validity of the findings

a. Despite limitations implied by retrospective data, the authors presented good evidence about the use of surgery in selected cases of stage IV breast cancer. It can help in directing future studies.
b. Unfortunately, the evidence coming from this study is not powerful enough to change the standard of care. However, it can provide another solid option to physicians and patients regarding the management.

Additional comments

The well-written manuscript, raw data, statistical analysis done and the presented results indicate how much effort and time the authors have spent in this study.

Although the data is retrospective with several limitations as the authors themselves described, the study added new bricks to the growing evidence in the matter of resecting oligometastatic disease.

Still, the study failed to characterize and examine the effect of different treatment approaches, such as new anti-her2 or endocrine therapies. Moreover, the study failed to identify the timing effect of local treatment and the best modality to deal with it. Unfortunately, there is no head to head comparison between radiotherapy and surgery in this issue, especially for a single metastasis. When it comes to defining patients' groups; Is there a cut-off for the number of metastases, it is three, four, or just one. This study failed to answer all of these questions. We are looking forward to seeing more solid RCTs to answer these questions.

---

## Round 0.2 · Minor Revisions

I would like to thank the reviewer for his thoughtful re-reading of this manuscript. Please add a point-by-point reply to the reviewer's comments. I agree with this evaluation and I would, therefore, request for the manuscript to be revised accordingly.

·

Basic reporting

I congratulate the authors for implementing the suggestions in their manuscript. I believe the manuscript is now concise and prepared for publishing.


A) This should take the authors only a few minutes to correct:

1. At the end of the second paragraph in discussion: »However, other experimental studies indicated that local surgery could substantially reduce overall tumor burden and improved immunologic response to cancer by inducing immune suppression and restoring immunocompetence [22-23].«
a. Please see the comment in the first review again. This statement is still incorrect. I would suggest writing something like: “However, other experimental studies in the murine model indicated that substantially reducing the overall tumor burden with local surgery leads to improved survival. This could be explained by the primary tumor’s suppression of the immune response and its removal leading to restored immunocompetence.”

2. Figure 4c: Replace "Solitary bone metastasis" with "bone-only metastases".

3. Table 4: Replace "Solitary bone metastasis" with "bone-only metastases", in two places.


B) This might as well be my misunderstanding of the presented data. I only ask the authors to recheck the numbers before publishing to be sure everything is in order.
According to figure 1 there were 13034 cases before propensity score matching and 4538 after (2269 in each of the breast-surgery and no-surgery group). However, the numbers in the presented tables don’t add up.

i. Table 2. All the numbers in the table should be rechecked.
1. The n numbers should be the same for BCSS and OS as these were analysed in the same group of patients.
2. The numbers do not add up. For example, the sum of the two highlighted numbers (see the pdf attachment) is 7433 below BCSS and 7949 below OS, but there should be 13034 cases according to figure 1.

ii. Table 3. All the numbers in the table should be rechecked.
1. The n numbers should be the same for BCSS and OS as these were analysed in the same group of patients.
2. The numbers do not add up. For example, the sum of the two highlighted numbers (see the pdf attachment) is 2438 below BCSS and 2603 below OS, but there should be 4538 cases according to figure 1.

iii. Table 5. All the numbers in the table should be rechecked. According to the manuscript there should be 2269 patients in the breast-surgery group after propensity score matching.
1. The n numbers should be the same for BCSS and OS as these were analysed in the same group of patients.
2. These numbers do not add up. For example, the sum of the two highlighted numbers (see the pdf attachment) is 1086 below BCSS and 1171 below OS, but there should be 2269 cases.

Experimental design

Please see 1. Basic reporting.

Validity of the findings

Please see 1. Basic reporting.

Additional comments

Please see 1. Basic reporting.

---

## Round 0.3 · accepted · Accept

The authors addressed the reviewers' concerns and substantially improved the content of MS. So, based on my own assessment as an editor, no further revisions are required and the MS can be accepted in its current form.

·

Basic reporting

no comment

Experimental design

no comment

Validity of the findings

no comment

Additional comments

Manuscript title: Impact of surgical treatment on survival for patients with stage IV breast cancer: a population-based propensity score matching analysis (#41899)

Third review

This retrospective study analysed impact of breast surgery on outcomes in de novo metastatic breast cancer patients using SEER derived data. The authors implemented the suggested corrections of the previous reviews well. The research question is relevant, the methodology is sound, and the results are well interpreted. I congratulate the authors for their manuscript, and I recommend this paper for publishing.